# An interferon-stimulated long non-coding RNA *USP30-AS1* as an immune modulator in influenza A virus infection

Yi Cao[1], Alex W. H. Chin[1,2], Haogao Gu[1], Mengting Li[1], Yuner Gu[1], Sylvia P. N. Lau[1], Kenrie P. Y. Hui[1,2], Michael C. W. Chan[1,2], Leo L. M. Poon[1,2,3,4]*

**1** School of Public Health, Li Ka Shing Faculty of Medicine, The University of Hong Kong, Hong Kong SAR, China, **2** Centre for Immunology & Infection, Hong Kong Science Park, Hong Kong SAR, China, **3** HKU-Pasteur Research Pole, The University of Hong Kong, Hong Kong SAR, China, **4** HKJC Global Health Institute, The University of Hong Kong, Hong Kong SAR, China

\* llmpoon@hku.hk

**Data Availability Statement:** RNASeq data can be available from GitHub (https://github.com/Leo-Poon-Lab/USP30-AS1-regulates-antiviral-immunity/). All other relevant data are within the

## Abstract

Long non-coding RNAs (lncRNAs) are essential components of innate immunity, maintaining the functionality of immune systems that control virus infection. However, how lncRNAs engage immune responses during influenza A virus (IAV) infection remains unclear. Here, we show that lncRNA *USP30-AS1* is up-regulated by infection of multiple different IAV subtypes and is required for tuning inflammatory and antiviral response in IAV infection. Genetically inactivation of *USP30-AS1* enhances viral protein synthesis and viral growth. *USP30-AS1* is an interferon-stimulated gene, and the induction of *USP30-AS1* can be achieved by JAK-STAT mediated signaling activation. The immune regulation of *USP30-AS1* is independent of its proximal protein-coding gene *USP30*. In IAV infection, deletion of *USP30-AS1* unleashes high systemic inflammatory responses involving a broad range of pro-inflammatory factors, suggesting *USP30-AS1* as a critical modulator of immune responses in IAV infection. Furthermore, we established a database providing well-annotated host gene expression profiles IAV infection or immune stimulation.

## Author summary

Influenza A virus (IAV) infection can induce differential expression of long non-coding RNAs (lncRNAs). However, the understanding of IAV induced lncRNAs that involve in host immune responses is limited. Here we identified that lncRNA *USP30-AS1* was induced by multiple subtypes of IAV infection, serving as a critical regulator regarding IAV induced immune responses. Deletion of *USP30-AS1* led to enhanced viral protein synthesis and elevated viral growth in IAV infection. JAK-STAT signaling activation can drive the transcription of *USP30-AS1*. *USP30-AS1* does not exert function through the nearby partner protein-coding gene *USP30*. Deficiency of *USP30-AS1* triggers unbalanced, elevated pro-inflammatory responses in IAV infected cells, indicating the role of *USP30-AS1* as a modulator regarding immune response during IAV infection. We also

main manuscript and Supporting information files
(Data S1–S8).

**Funding:** This work was supported by the Theme-based Research Scheme of the Research Grants Council, HKSAR Government (T11-712/19-N to M. C.W.C. and L.L.M.P.), the Health and Medical Research Fund, HKSAR Government (CID-HKU2 to L.L.M.P), the US National Institute of Allergy and Infectious Diseases, National Institutes of Health (HHSN272201400006C to M.C.W.C and L.L.M.P), and InnoHK, an initiative of the Innovation and Technology Commission, the Hong Kong Special Administrative Region (C2i to M.C.W.C and L.L.M. P). The funders had no role in study design, data collection and analysis, decision to publish, or preparation of the manuscript.

**Competing interests:** The authors have declared that no competing interests exist.

provide a user-friendly database that allows access to well-annotated host gene expression profiles in influenza virus infection or interferon stimulation.

## Background

Influenza A virus (IAV) is a segmented, single-strand RNA (ssRNA) virus of medical importance [1]. It can be classified into different subtypes based on the form of viral membrane glycoprotein hemagglutinin and neuraminidase [2]. During infection, IAV processes the cycle of infection to propagate virus progeny, and which involves series of virological and biological actions, including the attachment and entry of viral particles to the host cells, viral RNA transcription and replication, viral protein synthesis, virion assembly and budding [3].

Upon infection, IAV viral genomes are released from internalized viral particles into intracellular space, and are recognized by multiple host innate sensor machinery located in cytosolic or endosomal membranes. Retinoic Acid-inducible Gene-I like receptors (RLRs) and Toll-Like Receptors (TLRs) are major responders to IAV viral ssRNA [4–7]. By sensing viral RNA, their mediated signaling is transduced to the downstream through either RLRs-MAVS-TBK/IKKε axis or TLR7/8-MyD88-dependent signaling pathway, by which triggered cascade responses activate the transcription factor (TF) families of antiviral immunity and inflammation, including NFκb and multiple IRFs, promoting the transcription of interferon genes [8–10]. As a subsequent reaction, secreted extracellular interferons mediate the activation of JAK-STAT signaling for reinforcing and amplifying antiviral immunity, in which a series of signaling transductions are engaged. The interaction between interferons and their receptors induces tyrosine phosphorylation of receptor anchored kinase proteins, such as JAK1 and TYK2, which serve as hub for energizing STAT TF family, particularly STAT1 and STAT2, and shapes them into different active forms. This allows different combinations of phosphorylated STAT1 and STAT2 to bind to Interferon Sensitive Response Elements (ISREs), a short genomic DNA sequence motif that controls the activation of proximal immune genes. STAT1/2-ISREs binding contributes to the production of multiple major antiviral effector families, such as ISG, MX and OAS family, therefore performing JAK-STAT signaling mediated anti-viral and inflammatory responses [8,11–16]. Besides, viral sensor signaling can also directly induce the transcriptional activation of a range of pro-inflammatory cytokine genes, as one of the primary responses to the infection [17–19]. Importantly, treating cells with interferon, in turn, can also strengthen virus sensor signaling [20], indicating a potential positive feedback loop in response to interferon, and this amplified immune reaction is thought to be beneficial to restrict virus infection. However, sepsis or high systemic inflammatory responses might be triggered when pro-inflammatory signaling is overreacted [21,22] and it was associated with delayed virus clearance [23] and increased viral replication [24] in infection of influenza viruses and SARS-CoV-2 [25–31], suggesting the potential existence of modulators that balance inflammatory responses and other important biological processes.

Long non-coding RNAs (lncRNAs) regulate various biological activities through multiple mechanisms, in the level of gene transcription, post-transcriptional modification, and genome structure organization [32]. As reported by different loss-of-function or perturbation experimental systems, lncRNAs influence coding capacities of diverse groups of genes and their functions, including immune effectors [33–36]. Interestingly, influenza virus infection alters the profile of numerous lncRNAs, in which a sizable lncRNAs is likely triggered by interferon signaling [37–39]. Individual investigations of lncRNAs demonstrate their roles in modulating

key antiviral effectors such as *MX1* and *OASL*, rewiring antiviral pathways or regulatory networks during infection [40,41].

Yet, an important scientific question about how lncRNAs engage immune responses in the context of IAV infection, remains unclear. Here, by performing bioinformatic analyses, together with a series of virological and biological assays, we identified lncRNA *USP30-AS1* as a crucial immune regulator in IAV infection.

## Results

### Differentially expressed lncRNAs in infection of multiple IAV subtypes

IAV infection can induce up-regulation or down-regulation in a variety of host genes of different biotypes, including lncRNAs. However, most universally differentially expressed lncRNAs across infections by different IAV subtypes remain to be identified. To explore the question, we constructed a working pipeline (S1 Fig). We searched public databases for datasets generated from different subtypes of IAV infection in hosts of human origin, including organoid, primary cells, or cell lines. In total, high-throughput datasets produced from bulk RNA-seq or microarray containing ten different IAV subtypes infections at single or multiple time points were selected [42–47] (Table 1).

By analyzing the transcriptomic data, in comparison with mock infection, we identified in total 1715 differentially expressed lncRNAs (fold change > 1.5, adjusted *P*-value < 0.05) in at least one post-infection time point (S1 Table). The side bars and their annotated number of Fig 1A showed the distribution of these 1715 lncRNAs across infection datasets from different IAV subtypes. Due to variation introduced from infection quality, library preparation and high-throughput platform difference, detected number of differentially expressed host lncRNAs varies from 592 in A/Wyoming/03/03 (H3N2) infection dataset to 44 in dataset of A/Netherlands/602/2009 (H1N1) infection. In Fig 1A, lncRNAs differentially expressed in multiple different IAV infection datasets were summarized in the main bar, in which the above indicated number represents the counts of different individual lncRNA(s) that were differentially expressed in black dot indicated infection datasets in the dot matrix. Based on the counts, 65 out of 1715 lncRNAs were differentially expressed in infection of at least five different IAV subtypes (Fig 1A, S1 Table). Of the 65 lncRNAs, *USP30-AS1* and *IRF1-AS1* were the most universally up-regulated long non-coding transcripts spanning infection by eight different IAV subtypes, followed by a panel of eight

**Table 1. Included high-throughput datasets from hosts infection experiments in the study.**

| Origin BioProject | Platform type | Host | IAV subtype |
|---|---|---|---|
| PRJNA557257 | RNA-seq | Lung explants | A/Panama/2007/1999 (H3N2) |
| PRJNA795161 | RNA-seq | Human bronchial epithelial cells (HBEC) | A/Oklahoma/447/2008 (H1N1) |
| PRJNA667475 | RNA-seq | A549 | A/WSN/1933 (H1N1) |
| PRJNA349748 | RNA-seq | Human tracheobronchial epithelial cells (HTBE) | A/California/04/09 (H1N1); A/Wyoming/03/03 (H3N2); A/Vietnam/1203/04 (H5N1) HALo |
| PRJNA382632 | RNA-seq | Human monocyte-derived macrophages (MDM) | A/California/04/09 (H1N1); A/Wyoming/03/03 (H3N2); A/Vietnam/1203/04 (H5N1) HALo |
| PRJNA139513 | Microarray | Calu-3 | A/Vietnam/1203/2004(H5N1) |
| PRJNA163315 | Microarray | Calu-3 | A/California/04/09 (H1N1) |
| PRJNA175069 | Microarray | Calu-3 | A/Netherlands/602/2009 (H1N1); A/California/04/09 (H1N1) |
| PRJNA215073 | Microarray | Calu-3 | A/Anhui/01/2013 (H7N9); A/Netherland/219/2003 (H7N7); A/Vietnam/1203/2004 (H5N1); A/Panama/2007/1999 (H3N2) |

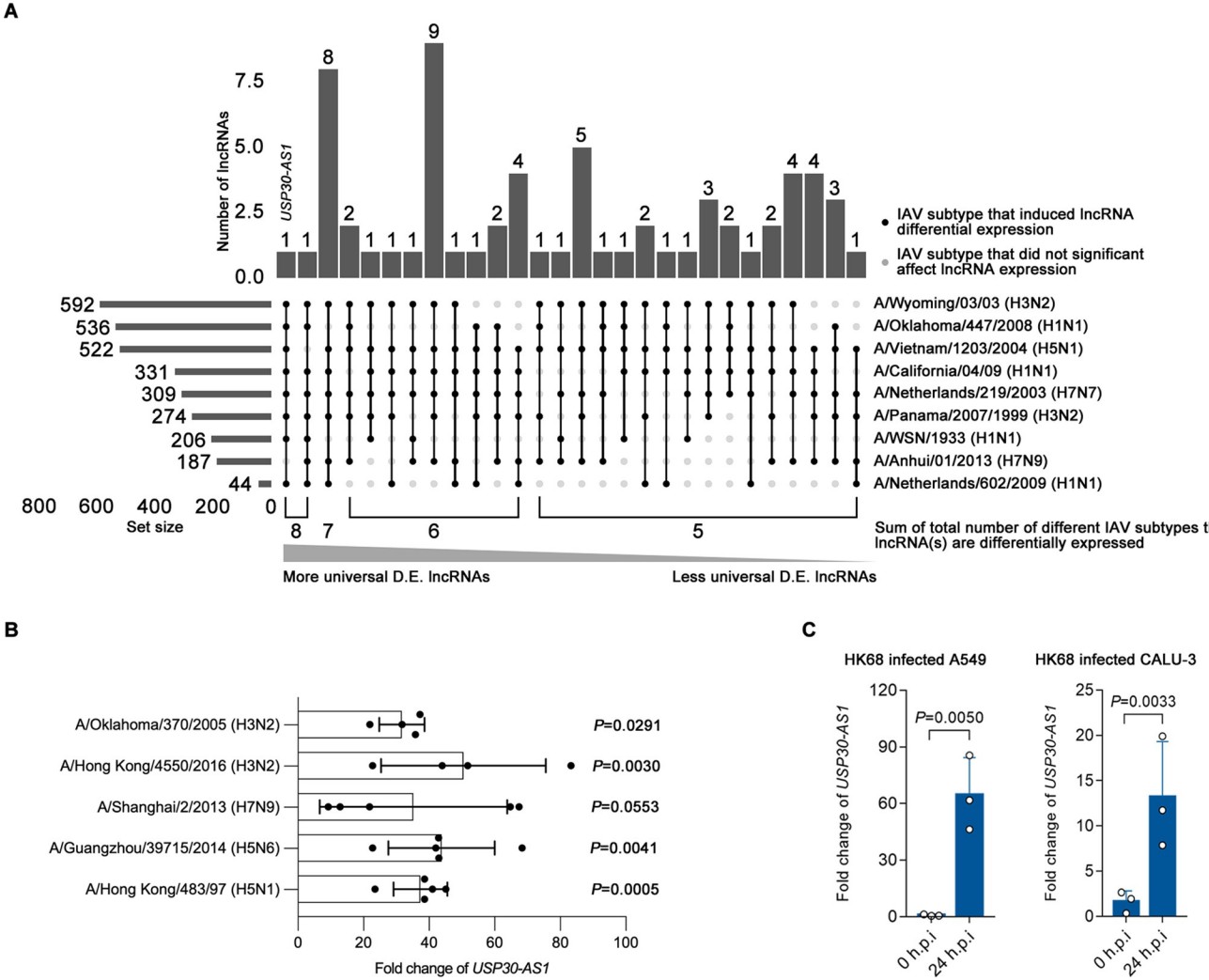

**Fig 1. Most universally differential expressed lncRNAs across infection of different IAV subtypes.** A) Upset plot showing the distribution of lncRNAs that were differentially expressed in infection of at least 5 different IAV subtypes in included datasets in the study. The main bar and above number represent the sum of individual different lncRNAs that differentially expressed in black dot indicated infection datasets. The side bar and annotated number indicate the total number of differentially expressed lncRNAs in each infection dataset. B) The expression fold change of *USP30-AS1* in A/Hong Kong/483/97 (H5N1), A/Guangzhou/39715/2014 (H5N6), A/Shanghai/2/2013 (H7N9), A/Hong Kong/4550/2016 (H3N2) or A/Oklahoma/370/2005 (H3N2) infected human primary alveolar epithelial cells (with M.O.I. of 2) compared to mock infected control at 24 hours post infection (h.p.i.). Two independent experiments were conducted, a representative experiment was shown. Student's t-test was used to test the difference between infection and mock. The bar height represents mean and error bar represents standard deviation. C) Expression fold change of *USP30-AS1* expression in A/Hong Kong/1/68 (H3N2) (M.O.I. of 0.1) infected A549 or CALU-3 versus PBS treated A549 or CALU-3 at 0 and 24 hours post-infection (h.p.i.). Experiment was conducted in triplicates. Student's t-test was used to test the difference between infection and mock group. Significant exact two-sided *P*-value was reported. The bar height represents mean and error bar represents standard deviation.

lncRNAs (*BCAR4*, *LINC02880*, *RFPL3S*, *NEXN-AS1*, *LINC00158*, *MIR155HG*, *LINC01127*, *LOC646626*) that were differentially expressed in infection by seven different IAV subtypes. Because A/Vietnam/1203/04 (H5N1) HALo is a vaccine strain [48], it was excluded in this part of analyses.

To study the transcriptional profiles trajectory of these 65 mostly differentially expressed lncRNAs, their expression across multiple time points were extracted, and datasets with less than two time points during IAV infection were not included in this analysis. In addition, we

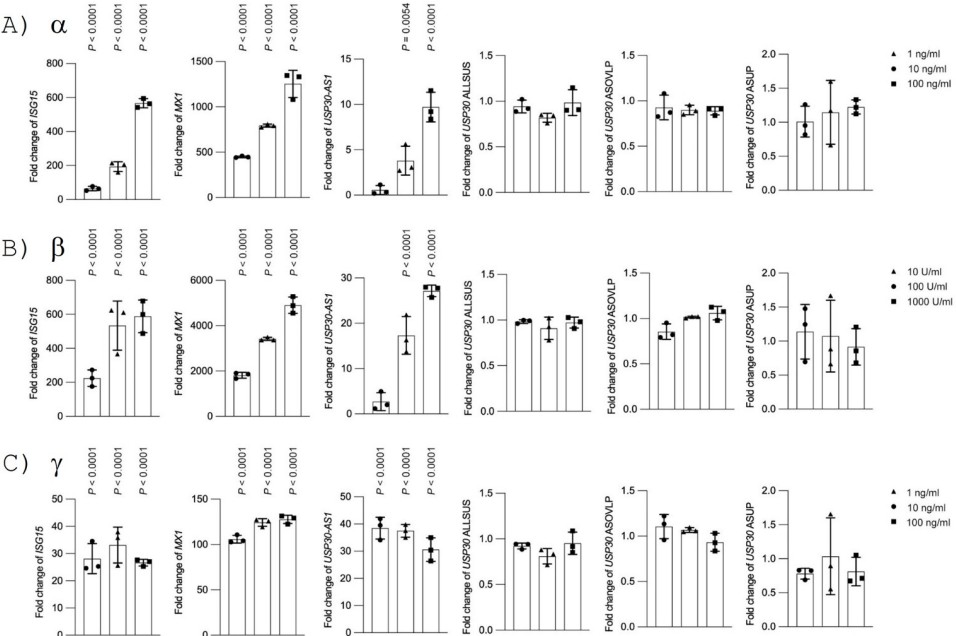

**Fig 2. The expression of *USP30-AS1* and different loci of *USP30* in response to interferon stimulation.** A) The expression fold change of *USP30-AS1* and different regions of *USP30*, as well as *MX1* and *ISG15* as positive controls, in response to different dose of interferon α stimulation compared to PBS-based mock treatment at 6 hours post-treatment. B) The expression fold change of *USP30-AS1* and different regions of *USP30*, as well as *MX1* and *ISG15* as positive controls, in response to different dose of interferon β stimulation compared to PBS-based mock treatment at 6 hours post-treatment. C) The expression fold change of *USP30-AS1* and different regions of *USP30*, as well as *MX1* and *ISG15* as positive controls, in response to different dose of interferon γ stimulation compared to PBS-based mock treatment at 6 hours post-treatment. *USP30* ALLSUS: *USP30* CDS region that are consensus across all different *USP30* transcripts; *USP30* ASOVLP: *USP30* genomic region that overlaps with *USP30-AS1*; *USP30* ASUP: *USP30* genomic region located in the upstream of *USP30-AS1*, overlapping with CDS of some transcripts of *USP30*. Experiment was conducted in triplicates. Student's t-test was used to test the difference between stimulation and mock group. Significant exact two-sided *P*-value was reported. The bar height represents mean and error bar represents standard deviation.

only kept time points less than 24 hours post infection to avoid the confounding effect from cell death. Among the 65 lncRNAs, most of them were over-presented, a few lncRNAs were downregulated, and the uptrend was positively associated with infection progression over time (S2 and S3 Figs). For these most universally differentially expressed lncRNAs, the trend of up-regulation was highly consistent across different IAV subtypes even when different host cells were used (S2 and S3 Figs). Together, our analyses suggest that some lncRNAs can be reproducibly induced in human cells by infection of different IAV subtypes.

## Characterization of *USP30-AS1*: An interferon-stimulated lncRNA

Of these highlighted lncRNAs, *USP30-AS1*, one of the most universally up-regulated lncRNAs during different IAV infection, caught our attention. Comparing to another highly ubiquitously induced lncRNA *IRF1-AS1*, which was reported functionally as pro-inflammatory factor and regulator of proximal gene *IRF1* [49,50], there is limited work on *USP30-AS1* and the understanding of this lncRNAs in IAV infection is completely lacking.

To further validate the induction of *USP30-AS1* by IAV, we tested the transcriptional activation of *USP30-AS1* in human primary alveolar epithelial cells isolated from non-malignant lung tissues of consented patients. Consistently, infecting alveolar epithelial cells with IAV remarkably

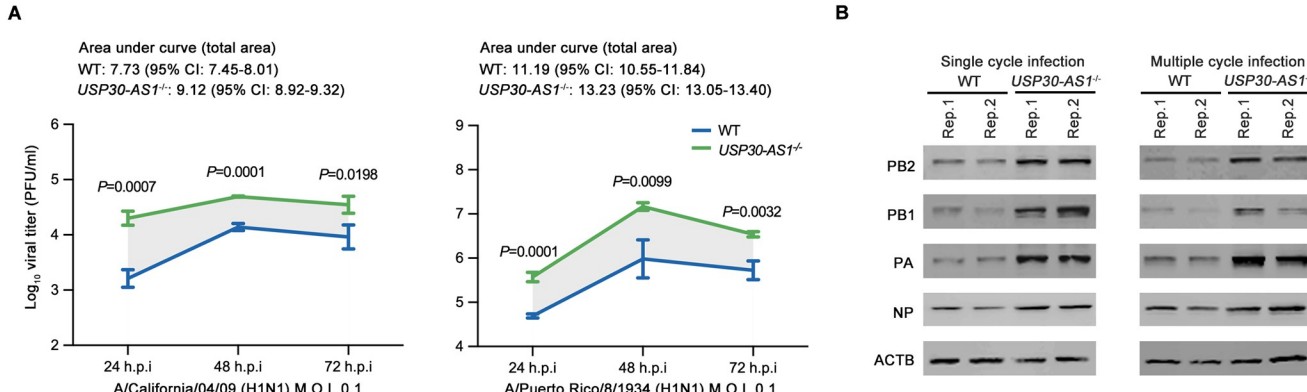

**Fig 3. Deletion of *USP30-AS1* strengthens IAV growth and promotes viral protein synthesis.** A) Viral titer in the supernatant of A/California/04/09 (H1N1) or A/Puerto Rico/8/1934 (H1N1) (M.O.I. of 0.1) infected *USP30-AS1*$^{-/-}$ A549 cells or WT A549 cells at 24, 48, and 72 hours post-infection (h.p.i.). A representative experiment was shown from two independent experiments performed in triplicates. Student's t-test was used to test the difference of viral titer in the supernatant between infected *USP30-AS1*$^{-/-}$ A549 and WT A549 cells. Significant exact two-sided *P*-value was reported. Mean and standard deviation were shown in the plot. CI, confidence interval. B) Detection of IAV viral PB2, PB1, PA, NP proteins and cellular ACTB protein expression by immunoblot in single cycle or multiple cycles A/Puerto Rico/8/1934 (H1N1) infected *USP30-AS1*$^{-/-}$ A549 cells compared to WT A549 cells.

increased the fold change of *USP30-AS1* in comparison with PBS treated mock control (Fig 1B). Besides, we also examined the expression of *USP30-AS1* in A/Hong Kong/1/1968 (H3N2) (HK68 in brief) infected A549 or CALU-3 cells and confirmed the elevated transcription level of *USP30-AS1* (Fig 1C). We then asked if *USP30-AS1* are interferon signaling dependent. We treated A549 cells with interferon α, β and γ of different doses and found that the stimulation of these treated types of interferon to A549 cells can enhance the expression of *USP30-AS1* (Fig 2A–2C; left 3 panels). We also quantified the transcriptional activity of nearby loci of *USP30-AS1* that contain different regions of its partner protein-coding gene *USP30*, and found these loci were not transcriptionally responsive to interferon stimulation (Fig 2A–2C, right 3 panels; S7A Fig). As interferon responsive genes have high frequency to carry ISREs motif in the promoter region, we scanned the motif pattern around *USP30-AS1* genome and identified two ISRE-like motifs (ACTTTCATTTTTA) in the upstream of *USP30-AS1* (S4C Fig), suggesting that a potential transcriptional regulation through TF binding.

As most of the lncRNA transcripts are poorly annotated, we performed RT-PCR reactions of 5' RACE, 3' RACE and head-to-tail exon-exon junction RT-PCR, together with nanopore long reads sequencing to identify the boundary and alternative transcripts of *USP30-AS1* in non-infected or IAV infected cells (S4A Fig). Nanopore-seq validated that the *USP30-AS1* expression was significantly upregulated during infection (S4B Fig). The identified transcription start site (TSS) of the dominant *USP30-AS1* transcript was slightly longer (around 19 bp) than the reported annotation. It is worth noting that the reads mapped to the TSS region of *USP30-AS1* had identical positions with the proximal ISRE-like motif (S4B and S4C Fig). It was also identified that the 3' end of *USP30-AS1* transcript was about 155 bp shorter than previously reported (S4B and S4C Fig). Besides, nanopore-RACE-seq discovered a novel transcript of *USP30-AS1* that was longer than the classical one. However, further experiments showed that it had relatively limited expression even under interferon stimulation or IAV infection (S4C and S4D Fig).

### Genetic inactivation of *USP30-AS1* enhances IAV viral protein synthesis and viral growth

To study the virological and biological effect of *USP30-AS1*, we generated a *USP30-AS1* full knock-out (KO) A549 cell line (S5A and S5B Fig). We first infected *USP30-AS1*$^{-/-}$ cells

or wide type (WT) cells with A/California/04/2009 (H1N1) (CA04 in brief) or A/Puerto Rico/8/1934 (H1N1) (PR8 in brief) and investigated virus growth kinetics. Viral titer in the supernatant of either CA04 or PR8 infected *USP30-AS1*$^{-/-}$ cells was remarkably higher than the viral titer in the supernatant of infected WT cells at all examined time points (Fig 3A).

Next, we performed a series of virological assays to determine whether *USP30-AS1* affects IAV infection cycle. We first tested if *USP30-AS1* influences virus internalization in CA04 or PR8 infection. Internalized viral genomes in *USP30-AS1*$^{-/-}$ cells were found to be comparable to that in WT cells, indicating that *USP30-AS1* does not affect viral entry (S5D Fig). To test the impact of *USP30-AS1* in IAV viral RNA transcription and replication, we performed single and multiple cycle infection of two H1N1 viruses. Quantification of vRNA, cRNA and mRNA showed that *USP30-AS1* had no significant effect on viral RNA transcription or replication in both single and multiple cycle infection (S5E and S5F Fig). However, immunoblotting data showed that viral protein synthesis was significantly enhanced in PR8 infected *USP30-AS1*$^{-/-}$ cells compared to infected WT cells in both single and multiple cycle infection (Fig 3B). Consistent with that, higher expression of viral proteins PA and NP were detected by immunochemical staining in PR8 infected *USP30-AS1*$^{-/-}$ cells in comparison with infected WT cells (S6A and S6B Fig). Together, those data suggested that the absence of *USP30-AS1* may enhance the production of viral protein and promote virus propagation during IAV infection.

## *USP30-AS1* affects IAV infection through *USP30* independent manner

As *USP30-AS1* is located in the opposite strand of *USP30*, it is possible that *USP30-AS1* might act as a regulatory element to modulate its partner protein-coding gene *USP30* during IAV infection. To test this hypothesis, we first investigated the expression of *USP30-AS1* and the consensus region of all *USP30* transcripts (S7A Fig) in single cycle CA04 infection with M.O.I. of 5. The expression of *USP30-AS1* was induced significantly over time, while the expression pattern of *USP30* was inconsistent (S7B Fig). As previous interferon stimulation experiments had shown that *USP30* did not respond to interferon (Fig 2A–2C) but considering that other secreted molecules during infection might play a role, we extracted conditioned medium, which is a virus-free cell culture medium containing a pool of diverse antiviral and inflammatory molecules secreted by infected cells, to treat A549 cells for determining how *USP30* would react to. With conditioned medium treatment, *USP30-AS1* was transcriptionally activated, while an inconsistent expression pattern of *USP30* was observed again (S7C Fig). These data suggest a still undefined relationship between *USP30-AS1* and *USP30*, and there's no apparent intrinsic covariation in their expression with one another, particularly in the context of IAV infection and the stimulation of inflammatory and antiviral molecules. This was also supported by the correlation analysis between expression of *USP30-AS1* and *USP30* in 1150 lung cancer-related tissues in the TCGA database (https://www.cancer.gov/ccg/research/genome-sequencing/tcga), in which no significant correlation was found between *USP30-AS1* and *USP30* (S7D Fig).

To test if *USP30* can affect IAV infection, we also generated *USP30*$^{-/-}$ A549 cell line (S5C Fig). Viral growth kinetics showed that no remarkable viral titer change in the supernatant was observed in CA04 or PR8 infected *USP30*$^{-/-}$ cells compared to infected WT cells (S7E Fig), indicating *USP30* may not significantly impact IAV infection. Collectively, we concluded that *USP30-AS1* may influence the process of IAV infection in a fashion independent of *USP30*.

## Transcriptional activation of *USP30-AS1* can be achieved via JAK-STAT signaling

Given that *USP30-AS1* can be triggered by interferons and conditioned medium, we hypothesized that the activation of *USP30-AS1* may be dependent on interferon-related signaling, such as JAK-STAT signaling pathway, a major signaling pathway that drives the expression of interferon stimulated genes (ISGs) upon activation by interferon. To test the hypothesis, we performed a series of assays blocking important signaling transducers in JAK-STAT by using different protein inhibitors. We first treated cells with JAK1 antagonists following interferon stimulation. Treated cells with JAK inhibitor I, an antagonist can impair the function of JAK1 and TYK2 blocking the signaling transduction therefore ablating the activation of STAT1/2 in the JAK-STAT pathway [51], dramatically abrogated the activation of *USP30-AS1* even under high concentration of Type I or Type II interferon stimulation (Fig 4A). We also treated cells with fludarabine [52], a STAT1 antagonist to impede all STAT1 related TFs complexes in a similar fashion. Treating cells with fludarabine showed reduced activation of *USP30-AS1* upon Type I, but not Type II, interferon stimulation (Fig 4B). Besides, we treated cells with Stattic [53], which is a STAT3 antagonist. It was observed that the activation of *USP30-AS1* was not affected by Stattic (Fig 4C).

To investigate whether other host defense immune signaling also plays a role in activation of *USP30-AS1*, we infected cells with either PR8 or CA04 with M.O.I. of 1 while treating cells with JAK inhibitor I. This allows cells to keep other antiviral immune pathways such as viral sensor signaling, without engagement of JAK-STAT signaling. Inhibition of JAK-STAT signaling during infection did not ablate the enhancement of *USP30-AS1* expression (Fig 4D). This might be because as a primary response, viral sensor signaling is also able to control the activation of a range of inflammatory cytokines, upon the recognition of viral components [17–19]. Considering the potential engagement of viral recognition pathways, we further studied if inhibiting core components in the viral sensor signaling can disrupt the induction of *USP30-AS1*. In low concentration, IKKε/TBK1 inhibitor II [54] is a potent antagonist that can inhibit IKKε and TBK1, a major signal transducer in RIG-I signaling, while in high concentration it has extended inhibitive effect on IKKα and IKKβ. Neither low nor high concentration of IKKε/TBK1 inhibitor II treated cells had impact on the activation of *USP30-AS1* in response to Type I or Type II interferon stimulation (Fig 4E) or IAV infection (Fig 4F). As viral components can be recognized by multiple sensor machinery, blocking individual transducer proteins solely might be not sufficiently enough to stop the whole signaling, and this might be the explanation of these results. Additionally, we also analyzed a published RNA-seq dataset generated from interferon-stimulated cells pre-treated with R848, a TLR7/8 agonist. No significant expression change was found in *USP30-AS1* (Fig 4G). Taken together, it suggests that JAK--STAT signaling is crucial for activating *USP30-AS1* in cells stimulated by interferons. However, the expression of *USP30-AS1* in IAV-infected cells can be triggered by mechanisms other than the JAK-STAT signaling pathway.

## Deficiency of *USP30-AS1* unleashes dysregulated high inflammatory response during IAV infection and immune stimulation

To parse which genetic programs are controlled by *USP30-AS1* during IAV infection, we performed bulk RNA-seq in CA04 or mock infected *USP30-AS1*<sup>-/-</sup> or WT A549 cells in triplicate. The samples were of good quality and high correlations were observed among the triplicates (S8A Fig). As different treatments and genotype of cells were used in the experiment, multifactor analysis was conducted. To directly identify *USP30-AS1* associated genes in IAV infection, the differentially expressed genes (DEGs) in CA04 infected *USP30-AS1*<sup>-/-</sup> cells compared

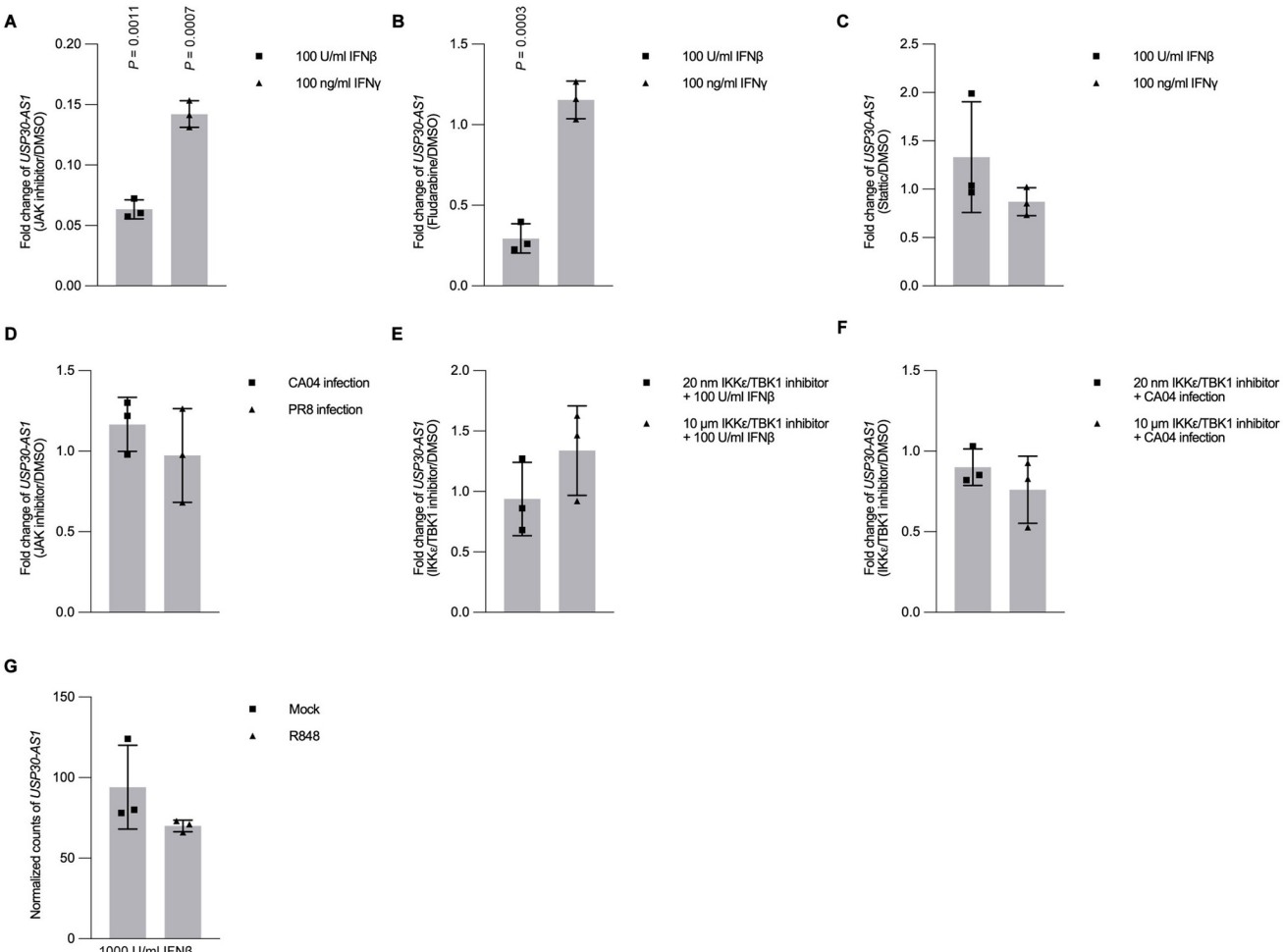

**Fig 4. Major host defense signaling and the expression of *USP30-AS1*.** A) Expression fold change of *USP30-AS1* between 15 nM JAK Inhibitor-I and DMSO pre-treated (for 24 hours) A549 cells in response to either 100 U/ml interferon β or 100 ng/ml interferon γ stimulation at 6 hours. B) Expression fold change of *USP30-AS1* between 5 μM Fludarabine and DMSO pre-treated (for 24 hours) A549 cells in response to either 100 U/ml interferon β or 100 ng/ml stimulation γ at 6 hours C) Expression fold change of *USP30-AS1* between 20 μM Stattic and DMSO pre-treated (for 24 hours) A549 cells in response to either 100 U/ml interferon β or 100 ng/ml interferon γ stimulation at 6 hours. D) Expression fold change of *USP30-AS1* between 15 nM JAK Inhibitor-I and DMSO pre-treated (for 24 hours) A549 cells infected with either A/California/04/09 (H1N1) (with M.O.I. of 1) or A/Puerto Rico/8/1934 (H1N1) (with M.O.I. of 1) at 24 hour post infection (h.p.i.). E) Expression fold change of *USP30-AS1* between either low concentration of 20 nM IKKε/TBK1 inhibitor II and DMSO, or high concentration of 10 μM IKKε/TBK1 inhibitor-II and DMSO pre-treated (for 24 hours) A549 cells in response to either 100 U/ml interferon β or 100 ng/ml interferon γ stimulation at 6 hours. F) Expression fold change of *USP30-AS1* between either low concentration of 20 nM IKKε/TBK1 inhibitor II and DMSO, or high concentration of 10 μM IKKε/TBK1 inhibitor-II and DMSO pre-treated (for 24 hours) A549 cells infected with A/California/04/09 (H1N1) (with M.O.I. of 5) at 6 hours post-infection (h.p.i.). G) Expression of *USP30-AS1* (normalized counts) in A549 cells in response to 1000 U/mL interferon β stimulation at 12 hours post stimulation after cells treated with either 1 μg/mL R848 or mock treatment (from publicly available datasets, BioProject PRJNA481248). Experiment was conducted in triplicates. Student's t-test was used to test the difference between two groups. Significant exact two-sided *P*-value was reported. The bar height represents mean and error bar represents standard deviation.

to CA04 infected WT cells were determined after statistical adjustment considering also non-infected samples (Fig 5A). Functional enrichment analysis of DEGs in infected *USP30-AS1*[-/-] cells revealed a genetic network in which gene clusters related to inflammatory processes were enhanced, while cellular biosynthesis was downregulated, compared to infected WT cells (S8C Fig).

Some genes may have changed their basal expression in the absence of *USP30-AS1* even in the uninfected normal cellular state and we aimed to identify genetic programs regulated by

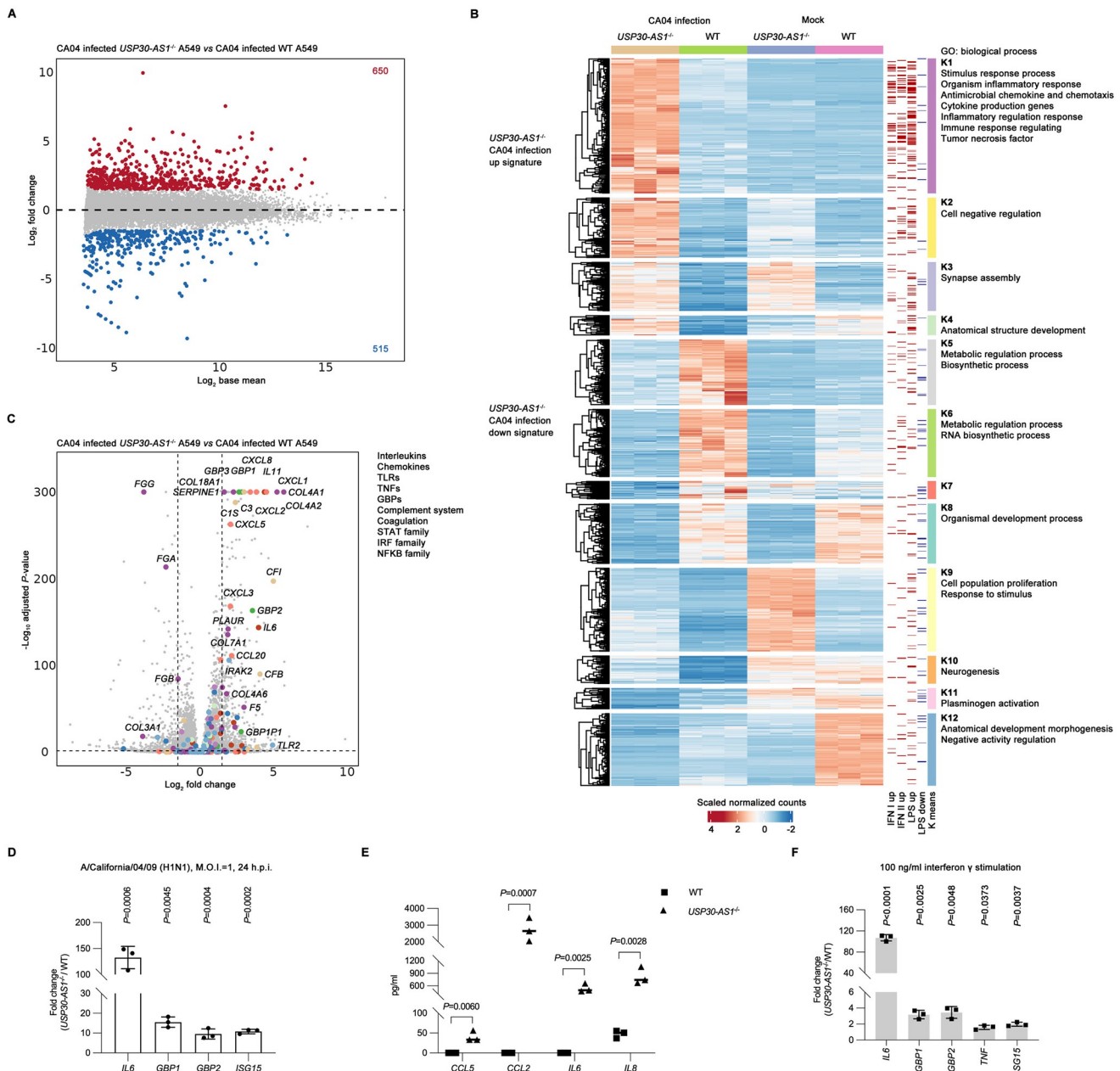

**Fig 5. Loss of *USP30-AS1* induces high systemic inflammatory response.** A) MA plot showing up-regulated genes (FC>1.5, adjusted *P*-value < 0.05) and down-regulated genes (FC<-1.5, adjusted *P*-value < 0.05) in bulk RNA-seq generated from either A/California/04/09 (H1N1) (with M.O.I. of 1) infected or mock treated *USP30-AS1*[-/-] A549 cells versus either infected or mock treated WT A549 cells. Differential expression calculation was performed under multi-factors adjustment. Bulk RNA-seq was performed in triplicates. B) Heatmap showing the K-mean clustered gene modules across cell genotype (*USP30-AS1*[-/-] or WT) and treatment (A/California/04/09 (H1N1) infection or mock infection) in the bulk RNA-seq data. Type I and Type II interferon stimulated signature genes, as well as up-regulated and down-regulated signature genes of LPS stimulation was also shown for comparison. Each clustered module was annotated with enriched biological processes in Gene Ontology (GO) analysis. C) Volcano plot showing the expression fold change of core members in families of major pro-inflammatory factors in A/California/04/09 (H1N1) infected *USP30-AS1*[-/-] A549 cells versus A/California/04/09 (H1N1) infected WT A549 cells after multi-factors adjustment from bulk RNA-seq data. D) Expression fold change (*USP30-AS1*[-/-] versus WT) of major pro-inflammatory factors observed in RNA-seq analysis in A/California/04/09 (H1N1) (with M.O.I. of 1) infected *USP30-AS1*[-/-] A549 cells compared to infected WT cells at 24 hours post-infection (h.p.i.), same condition performed as bulk RNA-seq. Experiment was conducted in triplicates. Student's t-test was used to test the difference between *USP30-AS1*[-/-] and WT during infection. Significant exact two-sided *P*-value was reported. Bar height represents mean and error bar represents standard deviation. E) Protein expression (pg/ml) of major pro-inflammatory mediators in supernatant from same experiment showed in D), by beads-based immunoassay. Experiment was conducted in triplicates. Student's t-test was used to test the difference between *USP30-AS1*[-/-] and WT group infection. Significant exact two-sided *P*-value was reported. Bar height represents mean and error bar represents standard deviation. F) The expression fold change of major pro-inflammatory molecules between *USP30-AS1*[-/-] A549 cells and WT cells in response to 100 ng/ml

interferon γ stimulation at 6 hours post-treatment. Experiment was conducted in triplicates. Student's t-test was used to test the fold change difference between *USP30-AS1*[-/-] and WT group infection. Significant exact two-sided *P*-value was reported. Bar height represents mean and error bar represents standard deviation.

*USP30-AS1* that are specifically impacted by IAV infection. To do this, we clustered the up-regulated and down-regulated genes across treatments and cell genotype by K-mean clustering and to explain at least 80% of the variation of the datasets, 12 clusters were applied (S8B Fig). The clustering revealed modules of *USP30-AS1* associated gene signatures in infection, which were exclusively induced or suppressed by IAV infection in the absence of *USP30-AS1* (Fig 5B). In line with previous data, the clusters (K1 and K2) that consist of the up-regulation signature of *USP30-AS1* in infection were mostly associated with inflammatory responses (Fig 5B). The *USP30-AS1* associated down-regulation signature (K5, K6, K7) during infection, too, in concordant with observed outcome, was associated with metabolic and biosynthetic processes (Fig 5B). These results suggest that *USP30-AS1* regulated modules account for the major transcriptional changes that control important biological processes in IAV infection. In the genetic background of *USP30-AS1* deficiency, IAV infection was able to potentiate stronger inflammatory responses and enhanced inhibition on cellular biosynthesis.

As multiple families of pro-inflammatory factors were involved in *USP30-AS1* associated genetic module (K1 cluster), we annotated core members in each pro-inflammatory family in a volcano plot of DEGs from IAV infected *USP30-AS1*[-/-] cells compared to infected WT cells. IAV infected *USP30-AS1*[-/-] cells had overall higher pro-inflammatory gene profiles than infected WT cells, including *IL6* in cytokine family, *GBP1*, *GBP2 and GBP3* in GBP family and various important numbers of chemokine family (Fig 5C). To validate this up-regulation, we quantified the expression of *IL6*, *GBP1*, *GBP2* and *ISG15* in the same infection condition as bulk RNA-seq by qPCR. Aligned with bulk RNA-seq results, the fold change of these genes was significantly higher in CA04 infected *USP30-AS1*[-/-] cells than infected WT cells during IAV infection (Fig 5D). In the same experiment, we performed a bead-based immunoassay that allows us to detect protein expression of multiple pro-inflammatory molecules in the supernatant. Consistently, the data showed the protein concentration of *IL6*, *IL8*, *CCL5* and *CCL2* was dramatically higher in supernatant of CA04 infected *USP30-AS1*[-/-] cells compared to supernatant of CA04 infected WT cells, suggesting an enhanced inflammatory response from both transcription and protein level (Fig 5E). Besides, stronger activation of complement system and coagulation was also found in infected *USP30-AS1*[-/-] cells in comparison with infected WT cells, which may contribute to high inflammatory responses as well (Fig 5C).

Interestingly, gene ontology analysis of overall up-regulated gene networks (S8C Fig) and up-regulated gene program in infected *USP30-AS1*[-/-] cells (K1) (Fig 5B) suggested a group of responses that are similar to bacterial infection, with *TLR2* up-regulation around 5-fold in CA04 infected *USP30-AS1*[-/-] cells compared to CA04 infected WT cells (Fig 5C). We then identified LPS stimulated genes in published datasets, and mapped them together with ISGs in the K-mean clustering heatmap. Consistently, LPS induced genes were preferentially enriched in the module of up-regulation in infection in the absence of *USP30-AS1* (K1 cluster), followed by genes triggered by Type II or Type I interferons (Fig 5B). This is consistent with a study that supports the bacterial infection-like gene molecular signature that contributes to manifestation of sepsis induced by severe SARS-CoV-2 infection [55].

With above observations, we ask whether this dysregulated high inflammatory response is due to increased sensitivity or responsiveness to infection-related stimulation, such as interferon γ, in *USP30-AS1*[-/-] cells. To test the notion, we treated *USP30-AS1*[-/-] and WT cells with

either interferon γ or PBS, and quantified cellular RNA expression of inflammatory cytokine genes. Indeed, in comparison with PBS treatment, a higher magnitude of fold change of *IL6*, *GBP1*, *GBP2*, *TNF* and *ISG15* was observed in interferon γ stimulated *USP30-AS1*$^{-/-}$ cells than in stimulated WT cells (Fig 5F). Having observed that interferon stimulation can trigger high inflammatory response in *USP30-AS1*$^{-/-}$, we also question whether the activation of viral sensor signaling can induce a similar effect. As expected, Poly (I:C) treatment also elicited stronger inflammatory and antiviral response in *USP30-AS1*$^{-/-}$ cells compared to WT, represented by augmented *IL6* and *ISG15* (S9 Fig), suggesting a possible intrinsic deficiency of controlling inflammatory and antiviral responses in reaction to the stress of IAV infection or other immune stimulation. Overall, these data suggest that *USP30-AS1* may engage genetic pro-grams that control inflammation and antiviral processes, and loss of *USP30-AS1* may lead to imbalanced high systemic inflammatory response upon IAV infection and other immune stimulations.

## Host Gene Expression under Influenza Virus Infection or Interferon Stimulation (GEII) database

To date, databases that provide easily accessible transcriptomic profile data in the context of IAV infection or interferon stimulation are still lacking. Considering this scientific need, we established a database named Host Gene Expression under Influenza Virus Infection or Inter-feron Stimulation (GEII) by integrating all high-throughput datasets included in the study. It shares well analyzed and annotated host gene profile data during influenza virus infection or interferon stimulation via a user-friendly interface. (https://leo-poon-lab-geii-scriptsweb-app-pk8r5m.streamlitapp.com/).

## Discussion

The role of lncRNAs in IAV induced immune responses has yet to be fully explored. By utiliz-ing published datasets, we first investigated the transcriptional profiles of lncRNAs in infection of multiple different IAV subtypes. We identified a group of universally differentially expressed lncRNAs across infection of ten IAV subtypes. Of these lncRNAs, *USP30-AS1* and *IRF1-AS1* were most ubiquitously up-regulated. We highlighted *USP30-AS1*, as there is limited work about this lncRNA in comparison to *IRF1-AS1*, which was relevant to interferon and NFκb signaling [49,50]. By performing stimulation experiments with different types of inter-ferons, *USP30-AS1* was identified as an interferon-stimulated lncRNA. For better investigating this lncRNA in detail, we performed experiments to characterize *USP30-AS1*. Through 5'/3' RACE coupled with long reads nanopore-seq, we identified the full length of *USP30-AS1* tran-scripts, and also found two ISREs motifs in the proximal upstream region of *USP30-AS1*, which might be serving as DNA binding sites upon TFs activation from infection-related signaling.

To determine if *USP30-AS1* affects IAV infection cycle, we performed a series of virological assays. It was found that knocking out of *USP30-AS1* enhanced viral protein synthesis without affecting IAV viral RNA transcription and replication, and overall, promoted viral growth. As *USP30-AS1* is an antisense RNA gene to the *USP30*, it is possible that *USP30-AS1* might carry out its biological function by regulating *USP30*. However, we found that the expression of *USP30* was independent of transcriptional activation of *USP30-AS1* and no significant virus growth was found in the supernatant of IAV infected *USP30*$^{-/-}$ cells compared to infected WT cells, suggesting *USP30* may not be a "hardwiring" responder of *USP30-AS1* in the context of interferon antiviral stimulation or IAV infection. However, the relationship between these two gene expressions in other cellular events might require further investigations.

As *USP30-AS1* can be induced by interferons, we investigated the interferon-related signaling pathways including JAK-STAT signaling, which is the primary antiviral response that activates inflammatory response and production of interferon. Treating cells with JAK Inhibitor can dramatically ablate the activation of *USP30-AS1* in response to Type I or Type II interferon stimulation. Blocking downstream TF STAT1 can impair the activation of *USP30-AS1* in response to Type I, but not Type II interferon stimulation. However, inhibiting JAK-STAT signaling does not significantly affect the activation of *USP30-AS1* during IAV infection, suggesting the potential engagement of other immune signaling. Considering this, we also inhibited multiple important transducer proteins in the viral sensor signaling, another primary immune defense pathway. Individually blocking transducer proteins of IKKε, TBK, IKKα and IKKβ in the viral recognition signaling also did not affect the activation of *USP30-AS1* significantly during IAV infection. This might be because *USP30-AS1* potentially responds to multiple viral sensing machinery, and inhibition of single transducer in the signaling might not be able to stop signaling transduction. Overall, above evidence supports *USP30-AS1* was driven by JAK-STAT signaling, while other major immune signaling such as viral recognition signaling might also involve the activation of *USP30-AS1*.

To understand the genetic programs that are influenced by *USP30-AS1* in IAV infection, we performed bulk RNA-seq in *USP30-AS1*[-/-] or WT A549 cells with either CA04 infection or PBS based mock infection. Gene clustering and gene ontology analysis revealed that *USP30-AS1* was associated with its specific genetic transcriptional modules, and which may control the processes of inflammation triggered by IAV infection. This high inflammatory response was validated in both transcriptional and protein levels. To test if this heightened immune response was specifically triggered by IAV, we also stimulated *USP30-AS1*[-/-] or WT cells with interferon γ or Poly (I:C). Interestingly, even without active viral replication engagement, such immune stimulation can also trigger elevated pro-inflammatory and antiviral responses, suggesting a potential dysregulation of immune reaction in response to infection or infection-related cellular stress. Overall, these data imply that lncRNA *USP30-AS1* may serve as a critical immune modulator in response to host defense.

There are a few limitations in the study. Firstly, the causal relationship between enhanced viral protein biosynthesis and high inflammatory responses induced by *USP30-AS1* deletion are not fully elucidated. Although we have identified that *USP30-AS1*[-/-] cells had increased sensitivity in infection-related stimulations that suggests an intrinsic deficiency of regulation of inflammatory responses, how IAV takes advantage of this immune dysregulation should be one of the important future works. Besides, one recent study indicates that *USP30-AS1* might be associated with regulation of SARS-CoV-2 infection [56], this raises an interesting question: whether other respiratory viruses can also induce the transcriptional activation of *USP30-AS1* and whether it plays a same role that modulate inflammatory response, and it should be further investigated. By leveraging other experimental tools, such as LNP-mRNA and cells defective in interferon response, extra investigation of *USP30-AS1* in virus infection and other cellular events is warranted. Furthermore, a more detailed examination of the subcellular locations of influenza viral proteins in infected *USP30-AS1*[-/-] cells could potentially uncover the mechanisms responsible for the increased virion production.

As the availability and accessibility of high-throughput data generated from influenza virus infection is still limited, by integrating all the datasets included in the study, we established a new database. The aim of the database is to help researchers generate hypotheses and validate experiment outcome by providing well annotated host gene profile data, including lncRNAs, through a user-friendly interface.

In conclusion, we identified an interferon-stimulated lncRNA *USP30-AS1*, which is one of the most universally induced lncRNAs by infection of multiple IAV subtypes, and it serves as a central modulator engaging immune responses in IAV infection.

## Methods

### Cells and viruses

A549, HEK-293T and MDCK cells were maintained in Minimum Essential Media (MEM) (11095080, Thermofisher) and supplemented with 10% heat inactivated fetal bovine serum (FBS) (16000044, Thermofisher) and 1% penicillin and streptomycin (P/S) (15140148, Thermofisher). Human primary alveolar epithelial cells were isolated from non-malignant lung tissues from consented patients, and were cultured in Small Airway Epithelial Cell Growth Medium (Lonza). All cells were kept in cell culture incubators at 37°C with 5% $CO_2$. A/California/04/09 (H1N1), A/Puerto Rico/8/1934 (H1N1) and A/Hong Kong/1/1968 (H3N2) were rescued from reverse genetics [57]. For culturing virus in chicken eggs, $1x10^3$ pfu/ml of influenza virus in 100 μl PBS were injected into 10-days embryonated eggs. Inoculated eggs were incubated at 37°C with 55–60% humidity for two days. Allantoic fluid from infected eggs was collected and centrifuged at 1000g at 4°C. Allantoic supernatant was aliquoted into 1.5 ml screw cap tubes and stored at -80°C freezer for further use. Viral titer of virus stock was determined by plaque assay. Viruses used for human primary cell infection, including A/Hong Kong/415742/2009 (H1N1), A/Oklahoma/370/2005 (H3N2), A/Shanghai/2/2013/ (H7N9), A/Guangzhou/39715/2014 (H5N6), A/Hong Kong/483/97 (H5N1), A/Hong Kong/4550/2016 (H3N2), were kindly donated from Dr. Michael Chan's lab.

### Plasmids and antibodies

Plasmids used for rescuing influenza viruses were prepared as previous described [57]. Plasmids for CRISPR-Cas9 gene editing (pSpCas9(BB)-2A-Puro (PX459) V2.0) were obtained from a nonprofit plasmid repository (62988, Addgene) [58].

Primary antibodies used in the study were USP30 (PA5-106762, thermofisher) (1:1000), PB2 (Sc17603, Santa Cruz Biotechnology) (1:1000), PB1 (NR31691, BEI) (1:1000), PA (PA5-31315, thermofisher) (1:1000 for immunoblot; 1:500 for IFA), NP (ab128193, Abcam) (1:2000 for immunoblot; 1:500 for IFA) and ACTB (Sc47778, Santa Cruz Biotechnology) (1:5000). Secondary antibodies used in the study were IRDye® 680RD Donkey anti-Goat IgG (H + L) (926–68074, LI-COR) (1:2000), IRDye® 680RD Donkey anti-Mouse IgG (H + L) (926–68072, LI-COR) (1:2000), IRDye® 680RD Donkey anti-Rabbit IgG (H + L) (926–68073, LI-COR) (1:2000) and Donkey anti-Mouse IgG (H+L) Highly Cross-Adsorbed Secondary Antibody Alexa Fluo 488 (A-21202, Thermofisher) (1:300).

### Preparation of knockout cells

A549 cells were seeded into wells of 24-well plates with 70% confluence. 500 ng PX459 plasmids were next diluted in Opti-MEM (31985070, Thermofisher) containing PLUS reagent as well as lipofectamine LTX reagent (15338100, Thermofisher) as instructed by the manufacturer. Plasmids-PLUS-LTX were then well mixed and incubated at room temperature for 5 mins. After that, the cell culture medium was replaced by Opti-MEM and the transfection mix was added into cells. Transfection medium was kept in cells for eight hours before being replaced by fresh culture medium. Cells were then treated with 2.5 μg/ml Puromycin for two days. Survived cells were individually seeded into wells in 96-wells for single cell clone

expansion. The identity of gene edited cell clones were verified by PCR, followed by DNA sequencing.

## RNA extraction and qRT-PCR

RNA samples from cells were extracted using RNeasy Mini Kit (74104, Qiagen) as instructed by the manufacturer. For detection of host gene expression, RNA was reverse transcribed using Super Script II Reverse Transcriptase (18064071, Thermofisher) with random primers. For detection of influenza viral vRNA, cRNA and mRNA, uni12 (5'-AGCAAAAGCAGG-3'), uni13 (5'-AGTAGAAACAAGG-3') and oligo(dT), respectively, were used in the corresponding reverse transcription reactions. Reverse transcription reactions were prepared as instructed by the manufacturer. PrimeSTAR GXL DNA Polymerase (R050A, Takara) was used for conventional PCR as instructed by the manufacturer. Fast SYBR Green Master Mix (4385612, Thermofisher) was used for qRT-PCR. $2^{-\Delta\Delta}$ CT values were introduced for quantifying gene expression. All the primers used in the research were listed in S2 Table.

## Influenza A virus infection

Human primary alveolar epithelial cells infection with different IAV subtypes, multiplicities of infection (M.O.I.) of 2 was used. Primary cells were incubated with serum-free MEM supplemented with 0.125 µg/mL TPCK-trypsin (4370285-1KT, Sigma) for one hour. After infectious incubation, cells were washed with PBS and replenished with serum-free medium. Cells were harvested 24 hour post infection.

For single-cycle IAV infection experiments, viruses were diluted in PBS to a predetermined concentration with M.O.I. of 5. Cells were washed twice with PBS and incubated with virus inoculum for one hour for absorption. Inoculum was then aspirated, and treated cells were washed twice with PBS. Infected cells were refilled with serum-free culture medium for cultivation. Infected cells were then harvested at indicated time points.

For detection of viral internalization, cells were infected with viruses of M.O.I. of 5 and incubated at 37˚C for one hour. After incubation, cells were first washed with acidified sodium chloride solution (pH 2) once and then washed with PBS once. Treated cells were cultured in serum-free culture medium at 37˚C for another half hour before harvesting.

For multiple-cycle IAV infection experiments, cells were infected by influenza virus (M.O.I of 0.1) using the procedures as described above. TPCK-trypsin was added to the cell culture medium (1 µg/ml for MDCK and HEK-293T cells; 0.5 µg/ml for A549 cells). Unless stated otherwise, infected cells were harvested at 24 hours post-infection for immunoblot and RNA-seq, and at 24, 48 and 72 hours post-infection for determining virus replication kinetics.

## Western blot

$1.76 \times 10^{5}$ cells were seeded into wells of 24-well plates a day before infection. After infection, supernatant was removed and cells were lysed in RIPA buffer (150mM NaCl, 1% NP-40, 5% DOC, 0.1% SDS and 50 mM Tris with pH 7.4) containing 1X Halt™ Protease Inhibitor Cocktail, EDTA-Free (87785, Thermofisher). Samples were then incubated at 4˚C with shaking for 10 mins. Lysates were centrifuged at 12000g at 4˚C for 10 mins to remove cell debris. Protein lysates were quantified by using BCA (23225, Thermofisher) method according to manufacturer's instructions. Appropriate volumes of 6X protein loading dye were added to the sample lysate to reach working concentration of 1X. Cell lysates with 1X protein loading dye were boiled at 95˚C for 10 mins. Protein loading volume was normalized to the sample with lowest protein concentration. SDS-PAGE gel with different percentages was used based on target protein's size. Nitrocellulose membrane was used for transferring proteins. Protein was blocked

by 3% BSA for half an hour. Primary antibody was added to the membrane and incubated with shaking overnight at 4°C. Secondary antibody was added to the membrane and incubated at room temperature in the dark for 30 mins. The membrane then was dried in the dark for one hour and scanned at Odyssey 9120 Near-Infrared Imager (LI-COR).

## Immunofluorescent assay

13-mm glass coverslips were washed twice with 70% ethanol for 5 mins each and then washed twice in pathogen-free PBS. Coverslip was coated with poly-L-lysine (P4832-50ML, Sigma) for 30 mins and then washed with PBS twice. A549 WT and *USP30-AS1$^{-/-}$* cells were seeded in wells of 24-well plates with 70% confluence before infection. Single-cycle infection experiment was conducted the next day, and infected cells were fixed at 4% paraformaldehyde (PFA) at six hours post-infection for at least one hour. Fixed cells were permeabilized in 0.2% triton X-100 at room temperature for 5 mins. Anti-PA (PA5-31315, Thermofisher) or anti-NP (ab128193, Abcam) monoclonal antibody was 500x diluted in TBS with 3% BSA and 30 μl of diluted antibody was added to the fixed cells. Coverslips were then kept in a dark, humidified chamber at 4°C overnight. Thirty μl 300x diluted secondary antibody (Anti-Mouse-Alexafluor 488) was added to treated cells the day after and incubated for half an hour in the dark. One drip of Pro-Long Diamond Antifade Mountant (P36965, Thermofisher) was added to stained cells. Mounted slides were air-dried for 30 mins and then then examined by fluorescence microscopy (Nikon ECLIPSE Ti-S).

## Plaque assay

MDCK cells were seeded to wells of 6-well plates and cultured to 100% confluence before conducting the plaque assay. Viruses were diluted from $10^{-1}$ to $10^{-6}$ with PBS. Cell culture medium was aspirated, and cells were washed once with PBS. Diluted viruses were then added to cells with 1 ml PBS in order and incubated at 37°C for one hour. Infectious liquid was then removed, and cells were washed with PBS once. 2 ml of mix of 1% melted SeaKem LE Agarose (50004, Lonza) with equal volume of culture medium containing 1 μg/ml TPCK-trypsin was added to cells. Cells were incubated for two days and then fixed with 10% formaldehyde for at least one hour. Fixed cells were stained with 1% crystal violet for half an hour. Stained plates were washed and dried for plaque counting.

## Interferon stimulation assay

Stimuli including interferon α2A (H6041-10UG, Sigma), interferon β (IF014, Sigma), interferon γ (GF305, Sigma), Poly (I:C) (tlrl-picw) and conditioned medium were used to stimulate cellular immune responses. Conditioned medium was generated from serum-free medium containing infected cells. Infectious serum-free medium was then collected and viruses in the medium were filtered away using Amicon Ultra-15 Centrifugal Filter Unit (UFC910024, Sigma). Filtrated medium containing cell secreted small immune-related proteins (e.g. interleukins, chemokines and interferons) was harvested for further use. For stimulation assay, cells were treated with the corresponding stimulus for 24 hours.

## Inhibition of innate immune signaling

For determining the upstream signaling of *USP30-AS1*, inhibitors to antagonize specific cellular signal transducers in the JAK-STAT signaling, viral recognition signaling pathways were used to treat cells before immune stimulation. JAK Inhibitor I (420099-500UG, Sigma), Fludarabine (HY-B0069, MCE), IKKε/TBK1 Inhibitor II (5063060001, Sigma) was used to treat

cells. Cells were treated with IKKε/TBK1 inhibitor II at a low concentration (20 nM) to antagonize IKKε and TBK1 or at a high concentration (10 μM) to inhibit IKKε and TBK1, together with IKKα and IKKβ. Cells were typically treated with the corresponding inhibitor for 24 hours, followed by interferon stimulation (100 U/ml of interferon β or 100 μg/ml of interferon γ) or IAV infection (M.O.I. of 5).

### Nanopore-RACE sequencing

Cells were seeded in T75 flasks in duplicate with 70% confluence a day before infection. Seeded cells were either infected with A/California/04/09 (H1N1) virus with M.O.I. of 1 or treated with PBS as mock infection for 24 hours. Infected cells were then lysed with Trizol (15596026, Thermofisher) and RNA was extracted as instructed by the manufacturer. The 5'/3' RACE was conducted based on the 5'/3' RACE System (18374–058;18373–019, Thermofisher). In brief, for 3' RACE RNA preparation, RNA was first heated to 65˚C for 5 mins and then quickly chilled on ice for 1 min. Chilled RNA was tailed with poly adenine by using *E. coli* Poly(A) Polymerase (M0276, NEB) based on manufacturer's instructions. RNA samples of 5' RACE and 3' RACE were then reverse transcribed by using SuperScrip IV Reverse Transcriptase (18090050, Thermofisher) with a gene specific primer (listed in S2 Table). For 5' RACE, reversely transcribed cDNA samples were subjected to cytosine-rich capping at the 5' end by using terminal deoxynucleotidyl transferase (EP0161, Thermofisher) with dCTP based on manufacturer's instructions. Next, the first round of PCR was conducted by using a gene specific primer paired with anchor primer (AAP) for 25 cycles. The PCR products were subjected to gel electrophoresis. If no target DNA band was presented in the gel, a nest PCR was performed using diluted first-round PCR product (5:495) with a primer pair containing nested gene specific primer and nested anchor primer (AUAP) for 30 cycles. For 3' RACE, polyA tailed RNA samples were reversely transcribed by using SuperScrip IV Reverse Transcriptase with a gene specific primer. Next, the first round of PCR was performed by using a gene specific primer plus a thymine-rich anchor primer (AP) for 25 cycles. The PCR products were subjected to gel electrophoresis. If no target DNA band was presented in the gel, a nest PCR was performed by using diluted first-round PCR product (5:495) with a primer pair containing nested gene specific primer and AP for 30 cycles. Exon-exon junction PCR was conducted by using a primer pair across the intron region of *USP30-AS1* for 40 cycles. For nanopore sequencing library preparation, all DNA products were subjected to gel purification by using Qiagen Gel Extraction Kit (28706X4, Qiagen). cDNA library was prepared by using Direct cDNA Sequencing Kit (SQK-DCS109, Nanopore) with Native Barcoding Expansion 1–12 (EXP-NBD104, Nanopore) according to manufacturer's instructions and was sequenced on Flow Cells at MinION sequencer for 72 hours.

### Bulk RNA sequencing

A549 WT and *USP30-AS1*[-/-] cells were seeded in T75 flasks in duplicate with 70% confluence a day before infection. Seeded cells were either infected with A/California/04/09 (H1N1) virus with M.O.I. of 1 or were treated with PBS-based mock infection for 24 hours. Infected cells were lysed with 4 ml Trizol and incubated at room temperature for 5 mins and 0.8 ml chloroform was then added to the lysate. The treated samples were centrifuged at 5000 g for 15 mins. Subsequently, 1 ml of liquid from the aqueous phase was carefully collected in a new tube and equal volume of RLT lysis buffer (from RNeasy Kit) was added to the samples. RNA was extracted according to the manufacturer's instructions. cDNA library of purified RNA was prepared by using KAPA mRNA HyperPrep Kit (08098140702, Roche). RNA samples that had passed QC were collected using poly-T oligo-attached magnetic beads to enrich mRNA

content. RNA was then fragmented to 200–300 bp, reverse transcribed using random hexamer, followed by second cDNA strand synthesis. After adaptor ligation, the libraries were enriched via 10 cycles of PCR. Illumina NovaSeq 6000 was used for Pair-End 151bp sequencing.

## Bead-Based immunoassays

Protein levels of human IL6, IL8, CXCL10 (IP-10), CCL2 (MCP-1), CCL5 (RANTES) and TNF in the culture supernatants were determined by the multiplexed, bead-based BD Cytometric Bead Array Flex Set (BD Bioscience) according to the manufacturer's instructions. In brief, culture supernatants and concentration standards were incubated with the capture bead mix for two hours, followed by incubation with the respective detection antibodies for one hour after washing. The bead samples were fixed with 4% paraformaldehyde, washed and analyzed using a BD LSR Fortessa Analyzer (BD Bioscience). Cytokine and chemokine concentrations were calculated with respect to the standard curves using FlowJo version 7.6.1 (BD Life Sciences).

## High-throughput datasets

Public datasets were obtained from Sequence Read Archive (SRA) or Gene Expression Omnibus (GEO) from NCBI or Influenza Research Database (IRD) from NIAID. Included datasets were listed in Table 1. For the RNA-seq dataset originated from NCBI BioProject PRJNA349748, PRJNA382632, processed differential expression data was downloaded from the Host Factor Experiment from Influenza Research Database (IRD). For the rest of the datasets, detailed analysis is described in the bioinformatic analysis part. Publicly available RNA-seq dataset from BioProject PRJNA481248 was obtained to investigate the expression of *USP30-AS1* in interferon stimulated cells after R848 treatment. Dataset from BioProject PRJNA795806 was used for identifying signature up-regulated or down-regulated genes in PBMC in response to LPS stimulation. TPM of *USP30-AS1* and *USP30* in 1150 lung cancer-related tissues were obtained from the TCGA database.

## Function annotation and enrichment analysis

Gene Ontology analysis was performed in g:Profiler (https://biit.cs.ut.ee/gprofiler/gost) [59]. To visualize the analysis, EnrichmentMAP [60] in Cytospace [61] was used. The enrichment map was built under type of generic, and GMT file downloaded from GO analysis in g:Profiler was used as input. FDR q-value cutoff was set as 0.001 and genes were filtered by expression. To reduce the redundancy, AutoAnnotate [62] was used to combine similar enriched biological terms and eventually a Collapsed AutoAnnotate enrichment map was generated.

## Bioinformatic analysis

For RNA-seq analysis, reads were mapped by using STAR aligner [63] (version 2.7.9a) with default parameters and mapped reads were counted by using featureCounts [64] (version 2.0.1) with default parameters except for pair or single end mode which was decided by sequencing type. Differential expression analysis was conducted by using R package DEseq2 (version 1.32.0). For nanopore sequencing, base calling was performed at HPC2021 System, HKU. Long reads were mapped by using minimap2 [65] (version 2.22) with default parameters and then visualized by using IGV genome browser.

## Statistical analysis

Student's t-test was used to test statistical difference between two groups and one-way ANOVA was used to test statistical difference beyond two groups, followed by *Tukey* post-hoc test, if multiple tests are needed. Two-sided significance level was applied in all statistical tests in the study.

## Supporting information

**S1 Fig. The working pipeline for high-throughput data generated from IAV infection.**
Schematic of working pipeline of database searching, selection criteria, included datasets and downstream analyses.
(TIF)

**S2 Fig. The expression kinetics of highly universally differentially expressed lncRNAs in included bulk RNA-seq datasets.** Heatmap showing the dynamic expression of lncRNAs that differentially expressed in infection of at least 5 different IAV subtypes in included bulk RNA-seq datasets.
(TIF)

**S3 Fig. The expression kinetics of highly universally differentially expressed lncRNAs in included microarray datasets.** Heatmap showing the dynamic expression of lncRNAs that differentially expressed in infection of at least 5 different IAV subtypes in included microarray datasets (selected example was presented if IAV subtype was included by various included datasets).
(TIF)

**S4 Fig. Full length and transcripts of USP30-AS1.** (A) Primers used for USP30-AS1 5' RACE, 3' RACE and exon-exon junction PCR, as well as primers for validating the results of exon-exon junction PCR. (B) Coverage (log scale) of mapped reads in USP30-AS1 genome from A/California/04/09 (H1N1) infected A549 cells or PBS mock infected A549 cells in duplicates. (C) Nanopore-RACE-seq determined TSS and TTS of USP30-AS1 with two ISREs in the upstream of USP30-AS1, and the detected novel transcript of USP30-AS1, as well as the possible RNA transcript splicing pattern. (D) Electrophoresis gel showing validated novel transcript of USP30-AS1.
(TIF)

**S5 Fig. Lack of USP30-AS1 promotes viral protein production.** (A) Diagram showing the genomic position of two sgRNAs targeting the upstream and downstream of USP30-AS1 at the same time. (B) Electrophoresis gel showing that compared to 2100 bp PCR product in WT A549 cells, the expected 133 bp PCR product was validated in USP30-AS1 full deletion cells by using detection primer pairs. (C) Immunoblot showing the USP30 protein expression in selected potential USP30 KO cell clones. KO#1 was picked and used for further IAV infection experiments. (D) Detection of expression fold change of IAV viral M gene between either A/California/04/09 (H1N1) or A/Puerto Rico/8/1934 (H1N1) internalized viruses in infected USP30-AS1$^{-/-}$ cells and infected WT cells by qPCR. Student's t-test was used to test the difference between two groups. Experiment was performed in triplicates. (E) Detection of IAV vRNA, cRNA and mRNA expression fold change of viral M gene between either A/California/04/09 (H1N1) or A/Puerto Rico/8/1934 (H1N1) single cycle infected (M.O.I. of 5) USP30-AS1$^{-/-}$ A549 cells and infected WT A549 cells at 6 hours post-infection (h.p.i.) by qPCR. The bar height represents mean and error bar represents standard deviation. (F) Detection of IAV vRNA, cRNA and mRNA expression fold change of viral M gene between either

A/California/04/09 (H1N1) or A/Puerto Rico/8/1934 (H1N1) multiple cycle infected (M.O.I. of 0.1) USP30-AS1$^{-/-}$ A549 cells and infected WT A549 cells at 24 hours post-infection (h.p.i.) by qPCR. Student's t-test was used to test the difference between two groups. Experiment was performed in triplicates. The bar height represents mean and error bar represents standard deviation.
(TIF)

**S6 Fig. Validation of inactivation of USP30-AS1 enhancing viral protein synthesis in IAV infection.** (A) Immunofluorescence of IAV viral protein PA in A/Puerto Rico/8/1934 (H1N1) infected USP30-AS1$^{-/-}$ A549 cells compared to infected A549 WT cells. (B) Immunofluorescence of IAV viral protein NP in A/Puerto Rico/8/1934 (H1N1) infected USP30-AS1$^{-/-}$ A549 cells or infected A549 WT cells. The nuclei were stained by DAPI (4',6-diamidino-2-phenylindole). Experiment was performed in triplicates.
(TIF)

**S7 Fig. The biological effect of USP30-AS1 is independent of USP30 in IAV infection or infection-related immune stimulation.** (A) Schematic showing the regions of RNA transcripts of USP30-AS1 and USP30 were detected for gene expression. (B) Detection of the expression fold change of USP30-AS1 (the left panel) and consensus regions of all USP30 transcript variants (the right panel) between single cycle of A/California/04/09 (H1N1) (M.O.I. of 5) infected A549 cells and mock infected A549 cells at 0, 4 and 8 hours post-infection (h.p.i.) by qRT-PCR. Student's t-test was used to test the difference between two groups in each time point. Significant exact two-sided P-value was reported. Experiment was conducted in triplicates. The bar height represents mean and error bar represents standard deviation. (C) Detection of the expression fold change of USP30-AS1 (the left panel) and consensus regions of all USP30 transcript variants (the right panel) between conditioned medium treated A549 and normal MEM medium treated A549 cells at 0, 2, 4, and 8 hours post treatment by qRT-PCR. Student's t-test was used to test the difference between two groups in each time point. Experiment was conducted in triplicates. The bar height represents mean and error bar represents standard deviation. (D) The correlation between Transcript Per Million (TPM) of USP30-AS1 and USP30 in 1150 in lung cancer-related tissues. Pearson correlation coefficient was calculated to test correlation coefficient between the two variables. (E) Growth kinetics in the supernatant of influenza virus A/California/04/09 (H1N1) (left) and A/Puerto Rico/8/1934 (H1N1) (right) infected USP30$^{-/-}$ A549 cells compared to infected WT A549 cells. Student's t-test was used to test the difference between two groups in each time point. Experiment was conducted in triplicates. The bar height represents mean and error bar represents standard deviation. N.s. indicates not significant.
(TIF)

**S8 Fig. Loss of USP30-AS1 triggers high systemic inflammatory response.** (A) Heatmap showing sample distance matrix across groups. (B) WSS methods to determine cluster number used in K-mean clustering. (C) Enrichment map showing collapsed biological processes networks of GO analysis in up-regulated genes or down-regulated genes in USP30-AS1$^{-/-}$ A549 cells versus WT A549 cells.
(TIF)

**S9 Fig. Poly (I:C) stimulation triggers a strong inflammatory response.** The expression fold change of pro-inflammatory cytokine, represented by IL6, and antivirals, represented by ISG15, between USP30-AS1$^{-/-}$ A549 cells and WT cells in response to 10 μg/ml Poly (I:C) stimulation. Experiment was conducted in triplicates. Student's t-test was used to test the fold change difference between USP30-AS1$^{-/-}$ and WT group. Significant exact two-sided P-value

was reported. Bar height represents mean and error bar represents standard deviation.
(TIF)

**S1 Table. Differentially expressed lncRNAs in infection of different IAV subtypes.**
(XLSX)

**S2 Table. Primers used in the study.**
(DOCX)

**S1 Data. Raw data for Fig 1.**
(XLSX)

**S2 Data. Raw data for Fig 2.**
(XLSX)

**S3 Data. Raw data for Fig 3.**
(XLSX)

**S4 Data. Raw data for Fig 4.**
(XLSX)

**S5 Data. Raw data for Fig 5.**
(XLSX)

**S6 Data. Raw data for S5 Fig.**
(XLSX)

**S7 Data. Raw data for S7 Fig.**
(XLSX)

**S8 Data. Raw data for S9 Fig.**
(XLSX)

## Acknowledgments

We acknowledge the research computing service provided by HPC, HKU. We also thank the contributors that generated published datasets included in the study.

## Author Contributions

**Conceptualization:** Yi Cao, Alex W. H. Chin, Leo L. M. Poon.

**Data curation:** Mengting Li.

**Formal analysis:** Yi Cao, Alex W. H. Chin, Haogao Gu, Yuner Gu, Sylvia P. N. Lau, Leo L. M. Poon.

**Funding acquisition:** Michael C. W. Chan, Leo L. M. Poon.

**Investigation:** Yi Cao, Yuner Gu, Sylvia P. N. Lau, Kenrie P. Y. Hui, Michael C. W. Chan.

**Methodology:** Yi Cao, Michael C. W. Chan.

**Project administration:** Leo L. M. Poon.

**Resources:** Kenrie P. Y. Hui.

**Software:** Haogao Gu, Mengting Li.

**Supervision:** Leo L. M. Poon.

**Writing – original draft:** Yi Cao, Alex W. H. Chin, Leo L. M. Poon.

**Writing – review & editing:** Yi Cao, Leo L. M. Poon.

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
