## [Decision Letter · Decision Letter 0]

15 Jul 2024

Dear Dr. Poon,

Thank you very much for submitting your manuscript "An Interferon-stimulated long non-coding RNA USP30-AS1 regulates antiviral immunity in influenza virus infection" for consideration at PLOS Pathogens. As with all papers reviewed by the journal, your manuscript was reviewed by members of the editorial board and by several independent reviewers. In light of the reviews (below this email), we would like to invite the resubmission of a significantly-revised version that takes into account the reviewers' comments.

We cannot make any decision about publication until we have seen the revised manuscript and your response to the reviewers' comments. Your revised manuscript is also likely to be sent to reviewers for further evaluation.

Sincerely,

Daniel R. Perez, PhD

Academic Editor

PLOS Pathogens

Kanta Subbarao

Section Editor

PLOS Pathogens

Michael Malim

Editor-in-Chief

PLOS Pathogens

orcid.org/0000-0002-7699-2064

Reviewer's Responses to Questions

**Part I - Summary**

Reviewer #1: The manuscript by Cao et al. addresses a relevant and novel aspect in the field of virus-host interactions. It studies the contribution of long non-coding RNAs to the host cell response against influenza infection.

Taking advantage of public RNAseq or microarray datasets (5 RNAseq and 4 microarray datasets) generated from human cells infected with influenza virus, the authors established a pipeline to identify differentially expressed lncRNAs common to infections with different influenza strains. Sixty-five lncRNAs were identified, with two of them being the most universally up regulated in cells infected with eight different influenza strains. Only one of them, USP30-AS1, was selected for further analysis.

Suppression of USP30-AS1 in knock-out (KO) A549 cells led to a significant increase in influenza viral titers, suggesting that USP30-AS1 had antiviral activity. Higher viral titers were associated with an increase in the levels of structural proteins with no significant effects in the amount of viral vRNA, cRNA and mRNA. RNAseq experiments showed an enhanced pro-inflammatory response in the absence of USP30-AS1 expression, suggesting an anti-inflammatory effect during infection with influenza virus.

The experimental results shown in the paper are promising and suggest a contribution of USP30-AS1 to the regulation of the inflammatory response in influenza virus infection. However, there are some concerns that should be addressed.

Reviewer #2: This is a very interesting study by Cao et al, investigating the expression of long non-coding RNA (lncRNA) in different cell types after influenza virus infection. They identified two lncRNAs that are upregulated in 100% of the studies: USP30-AS1 and IRF1-AS1. Following the extensive informatics analysis, they show that this RNA is upregulated ~60-fold and 12-fold in A549 and Calu-3 cells respectively after influenza virus infection (S5). Stimulation of A549 cells with type I or II interferons has a more modest effect on the expression of USP30-AS1 (Fig 2). Deletion of USP30-AS1 increased influenza virus titers in the supernatant of A549 cells and this increase was associated with higher viral protein levels in these cells. They also showed that inhibition of Jak-STAT and STAT pathway abrogated USP30-AS1 upregulation following type I or II interferon stimulation, but not influenza virus infection. Finally, they demonstrate in KO cells that deletion of USP30-AS1 is associated with increased expression of pro-inflammatory cytokine levels.

Overall, this is an interesting study on the role of lncRNA during influenza virus infection in vitro. The inclusion of the USP30 knockout cell data supports the overall conclusions that USP30-AS1 (and not USP30) inhibits influenza virus replication. Unfortunately, the authors did not assess the cytokine response in these cells to differentiate between potential direct effects of the lncRNA on USP30 expression and effects of USP30-AS1 on the virus and the host cell response. Ideally, the authors would have trans complemented the knockout cells with in vitro transcribed RNA (potentially packaged in LNP) to support their claims. Finally, it is unclear how USP30-AS1 controls protein expression after virus infection and how USP30-AS1 functions inside cells.

Major comments:

The title of the paper suggests USP30-AS1 is regulating antiviral immunity. What data support this conclusion? Based on the data presented, I see a model whereby the increase in virus protein production in the KO cells support higher levels of virus in the supernatant of infected cells. More virus and viral protein drive a stronger host cell response. An alternative model, based on a publication in Theranostics in 2022, implies that the loss of USP30-AS1 enhances the cellular response to virus infection, which in the case of influenza virus and A549 cells could be pro-viral. Demonstrating which of these two or other models is more likely would enhance the significance of the paper.

What is causing the difference in USP30-AS1 expression between virus infection and interferon stimulation? The variable data presentation (fold-change vs USP30-AS1/GAPDH ratio) makes it perhaps difficult to compare.

Minor comments:

Typo in Figure S5F…infetion.

Reviewer #3: Cao and colleagues investigated that role of long non-coding RNAs (lncRNAs) in influenza A virus infection (IAV). Employing publicly available datasets, they identify lncRNAs generally upregulated in the context of IAV infection. Among these was USP30-AS, which was subjected to detailed analyses. In brief, knock out of USP30-AS1 is shown to augment IAV infection, as determined by analyses of viral titers and expression of viral proteins in infected cells. Furthermore, evidence is provided that USP30-AS1 is upregulated by type I, II and III interferon (IFN) and that USP30-AS1 knock out results in a more pronounced upregulation of proinflammatory genes in the context of IAV infection. The results are of some interest to the field. However, important points remain open:

**Part II – Major Issues: Key Experiments Required for Acceptance**

Reviewer #1: 1. To further support the physiological relevance of lnc RNA USP30-AS1, it would be important to provide in vivo data confirming that influenza infection in vivo induces an increase in USP30-AS1.

2. Additional information for the interpretation of Fig. 1 should be provided in the text. Columns labeled with “1” apparently represent one specific lncRNA that is differentially expressed in several independent datasets. Do columns labeled with a different number indicate a set of differentially expressed lncRNAs?

3. Experimental results in Fig.S5 C, which validate experimentally the increased levels of USP30-AS1 in human cells infected with influenza virus, should be shown in the main manuscript, instead of in the supplementary material.

4. The methods for relative quantification of USP30-AS1, USP30 and other ISGs in Fig. 2 should be described more clearly. Was qPCR used to quantify these transcripts? ∆∆Ct method for relative quantification typically sets to 1 gene expression of the reference sample. In this case, the expression level in the absence of IFN stimulation would be 1 and the expression levels after IFN stimulation would provide a fold-change value. The values on the Y-axis, which start with “0” do not clearly show the fold-change in the expression level.

5. Fig. 3. Viral titers in the absence of USP30-AS1 were determined at low MOI (0.1), which corresponds to a multiple-cycle infection. Since protein and RNA levels were studied both at high and low MOI, viral titers should also be analyzed in a single-cycle infection (MOI 5).

6. The experimental design to determine the relationship between USP30-AS1 expression and the IFN response should be revised. The expression levels of USP30-AS1 are studied in non-infected cells that are treated with IFN (Fig. 4A, B and C). To study whether USP30-AS1 transcription is IFN-dependent, inhibition of IFN signaling pathway with STAT1/ JAK1 inhibitors should be performed in the context of infection. In fact, inhibition of IFN production in infected cells (Fig. E) did not have a significant effect on USP30-AS1 expression. Using cell lines or mutant viruses defective in the IFN response (either IFN production or IFN signaling) might be helpful to determine the physiological relevance of IFN in USP30-AS1 expression.

7. Fig. 5C. To further support the conclusion that USP30-AS1 contributes to down-regulate the inflammatory response during influenza virus infection, the expression of some relevant genes shown to be differentially expressed by RNAseq analysis should be validated by an alternative technique, such as quantitative PCR.

Reviewer #2: Do you see an increase HA and NA expression on the surface of KO cells. This would provide an opportunity to perform more mechanistic and kinetic studies to address the main critique of mechanism and model of this paper.

Attempt to trans-complement the KO cells with the lncRNA using transfection of mRNA-LNP.

Perform experiments that would differentiate between the models (see above).

Show that inhibition of JAK-STAT and STAT pathways inhibit USP30-AS1 expression in the context of influenza infection

Reviewer #3: USP30-AS1 upregulation in IAV infected respiratory epithelium cultured at the air-liquid interface should be shown.

The question how USP30-AS1 inhibits IAV infection remains inadequately addressed. Upregulation of pro-inflammatory genes should be confirmed on the protein level and it should be investigated how this upregulation impacts viral protein expression.

It should be investigated whether the impact of USP30-AS1 on inflammatory responses is IAV specific or also seen in the context of other respiratory infections.

**Part III – Minor Issues: Editorial and Data Presentation Modifications**

Reviewer #1: 1. The rationale to select USP30-AS1 over IRF1-AS1, also differentially expressed in infections with different influenza strains, should be provided.

2. Regarding the “replication” of the virus, these two statements might be apparently contradictory, please revise the wording: “USP30-AS1 had no significant effect on viral RNA transcription or replication” (lines 180-181)” and “the absence of USP30-AS1 may enhance the production of viral protein and promote virus replication”. In this last sentence, “virus replication” might be replaced by “virus growth”.

3. Table 1, which provides essential information about the cell lines used for the influenza infection experiments analyzed in the manuscript, should be moved from supplementary information to the main manuscript.

4. Fig. 3C is not mentioned in the manuscript. It could be kept in supplementary S5 Figure, since it is highly related to Fig.S5 G, which shows the expression of NP in USP30-AS1-/- cells infected with influenza virus.

Reviewer #2: The figures and order of presentation are a bit confusing. Why is fold increase in expression after HK68 infection presented in the supplement and not part of Figure 2? Similarly, what is the purpose of S5G? You have already shown this in Figure 3. Consider combining to enforce the point that protein expression is higher.

The role of S7 is unclear as it has limited relevance to this paper.

Reviewer #3: The authors should refer to influenza A viruses and not generally to influenza viruses.

It should be indicated for each figure subpanel, whether the results of a single representative experiment or the average of several experiments are shown. If a representative experiment is shown, please indicate the number of technical replicates and confirmatory experiments. If averages are shown, please indicate how many experiments were averaged. Further, please indicate whether error bars indicate SD or SEM.

PLOS authors have the option to publish the peer review history of their article (what does this mean?). If published, this will include your full peer review and any attached files.

Reviewer #1: No

Reviewer #2: No

Reviewer #3: No
---

## [Decision Letter · Decision Letter 1]

20 Dec 2024

Dear Dr. Poon,

We are pleased to inform you that your manuscript 'An Interferon-stimulated long non-coding RNA USP30-AS1 as an immune modulator in influenza A virus infection' has been provisionally accepted for publication in PLOS Pathogens.

Best regards,

Daniel R. Perez, PhD

Academic Editor

PLOS Pathogens

Kanta Subbarao

Section Editor

PLOS Pathogens

Michael Malim

Editor-in-Chief

PLOS Pathogens

orcid.org/0000-0002-7699-2064

Reviewer Comments (if any, and for reference):

Reviewer's Responses to Questions

**Part I - Summary**

Reviewer #1: The authors have satisfactorily addressed all the concerns raised by this reviewer.

The revised version of the manuscript includes additional experiments supporting the physiological relevance of long non-coding RNA USP30-AS1 in influenza A virus infection, which provide a more complete characterization of its function.

The discussion of the paper has also been significantly improved.

Reviewer #3: The authors have adequately addressed the points raised by this reviewer.

**Part II – Major Issues: Key Experiments Required for Acceptance**

Reviewer #1: (No Response)

Reviewer #3: (No Response)

**Part III – Minor Issues: Editorial and Data Presentation Modifications**

Reviewer #1: (No Response)

Reviewer #3: (No Response)

PLOS authors have the option to publish the peer review history of their article (what does this mean?). If published, this will include your full peer review and any attached files.

Reviewer #1: No

Reviewer #3: No

---

## [Editor Report · Acceptance letter]

30 Dec 2024

Dear Dr. Poon,

We are delighted to inform you that your manuscript, "An Interferon-stimulated long non-coding RNA USP30-AS1 as an immune modulator in influenza A virus infection," has been formally accepted for publication in PLOS Pathogens.

Best regards,

Sumita Bhaduri-McIntosh

Editor-in-Chief

PLOS Pathogens

orcid.org/0000-0003-2946-9497

Michael Malim

Editor-in-Chief

PLOS Pathogens

orcid.org/0000-0002-7699-2064